# Density-functional fluctuation theory of crowds

J. Felipe Méndez-Valderrama [1], Yunus A. Kinkhabwala [2], Jeffrey Silver [3], Itai Cohen[4] & T.A. Arias [4]

A primary goal of collective population behavior studies is to determine the rules governing crowd distributions in order to predict future behaviors in new environments. Current top-down modeling approaches describe, instead of predict, specific emergent behaviors, whereas bottom-up approaches must postulate, instead of directly determine, rules for individual behaviors. Here, we employ classical density functional theory (DFT) to quantify, directly from observations of local crowd density, the rules that predict mass behaviors under new circumstances. To demonstrate our theory-based, data-driven approach, we use a model crowd consisting of walking fruit flies and extract two functions that separately describe spatial and social preferences. The resulting theory accurately predicts experimental fly distributions in new environments and provides quantification of the crowd "mood". Should this approach generalize beyond milling crowds, it may find powerful applications in fields ranging from spatial ecology and active matter to demography and economics.

[1] Department of Physics, Universidad de Los Andes, Bogotá 111711, Colombia. [2] Department of Applied and Engineering Physics, Cornell University, Ithaca, NY 14853, USA. [3] Metron Inc., Scientific Solutions, Reston, VA 2019, USA. [4] Department of Physics, Cornell University, Ithaca, NY 14853, USA. These authors contributed equally: J. Felipe Méndez-Valderrama, Yunus A. Kinkhabwala  Correspondence and requests for materials should be addressed to T.A.A. (email: taa2@cornell.edu)

I dentifying the role of social interactions and environmental influences on living systems has been the goal of many recent studies of collective population behavior[1–15]. Current agent-based models of crowds can reproduce many emergent behaviors, ranging from random milling to swarming, but often must postulate preconceived rules for individual agent interactions with each other and their environment[1–10]. In contrast to such bottom-up approaches, some studies have inferred interaction rules from observations of individual motions within a crowd for a few species of fish[11,12], birds[13], and insects[14,15], but these studies have largely been limited to specific behaviors and have not been developed for making predictions under new circumstances. To date, a general predictive approach to emergent collective behavior in living systems has been lacking.

Such approaches, however, have been developed successfully for large collections of interacting atoms and molecules in the field of statistical physics. One of the central tenants of statistical physics is that generic thermodynamic behaviors emerge from underlying interaction rules among large numbers of particles[16,17]. Remarkably, these emergent behaviors are often insensitive to the detailed nature of the underlying interactions. Here, we pursue the hypothesis that a similar scenario emerges in the study of large crowds[18–22] so that behaviors arising from generic agent-based models can be predicted using a top-down approach. Accordingly, our strategy is to begin with a family of models that roughly capture the "microscopic" behaviors of individuals as they rearrange within a crowd. We do this, not because we are interested directly in individual behaviors, but rather because we are interested in the generic "macroscopic" behaviors that emerge in crowds en masse. This tack is not a priori obvious since active systems do not possess a fixed energy, their temperature is ill-defined, and there are no obvious equilibrium states[23]. Nonetheless, we show here that mathematical equivalents of free energy, the Hamiltonian, and equilibrium states arise naturally from plausible models of crowd behavior.

In this work, we present the following results. We introduce a general class of plausible agent-based models in which two different functions, "vexation" and "frustration," quantify location and social preferences, respectively. For this class of models, we develop a coarse-grained approach stemming from classical density-functional theory (DFT) that allows us to determine the general mathematical form of the probability distributions describing a crowd. We then discuss the conditions a system must possess to be describable by our theory and test our approach using a living system consisting of walking fruit flies (*Drosophila melanogaster*), which we confine to a variety of two-dimensional environments. For this fruit-fly system, we successfully extract the vexation and frustration functions corresponding to a variety of different physical settings. Furthermore, these functions are sufficiently stable that, by mixing and matching functions from different experiments, we accurately predict crowd distributions in new environments. Finally, by exposing the fly system to conditions that elicit distinct social motivations, we are able to identify changes in the overall behavior of the crowd, i.e., its "mood," by tracking the evolution of the social preference function.

## Results
**General mathematical form of crowd-density distributions.** Consider, as an example, a crowd at a political rally (Fig. 1a). Under such circumstances, individuals will seek the best locations —presumably closest to the stage—while avoiding overcrowded areas where there is insufficient "personal space." Moreover, individuals will, from time to time, move to new, better locations that become available.

A plausible agent-based model of this behavior would assign an intrinsic desirability of each location $\mathbf{x}$ through a "vexation" function $V(\mathbf{x})$ that takes its minimum value at the most ideal location near the stage. In addition, it would account for crowding effects through the local crowd areal density $n(\mathbf{x})$ by introducing a "frustration" function $f(n)$, so that the relative preferablity of location $\mathbf{x}$ is actually the sum of vexation and frustration effects, $V(\mathbf{x}) + f(n(\mathbf{x}))$. Finally, this model would include a behavioral rule to account for the tendency for individuals to seek improved locations. When an agent considers a move from location $\mathbf{x}$ to $\mathbf{x}'$, the change in the agent's dissatisfaction is $\Delta H \equiv (V(\mathbf{x}') + f(n(\mathbf{x}'))) - (V(\mathbf{x}) + f(n(\mathbf{x})))$. A rule where each agent executes such moves with probability $1/(e^{\Delta H} + 1)$ captures the intuition that moves that increase the dissatisfaction $\Delta H > 0$ are unlikely, and moves that decrease the dissatisfaction $\Delta H < 0$ are likely, while moves where $\Delta H = 0$ occur with 50% probability. The disadvantage of such an agent based modeling approach is that the rules for each agent are postulated and comparison with experiment requires gathering statistics from repeated simulations, each of which scales as the number of agents or worse. Again, our purpose here is not to develop such a model in detail, but rather to explore the top-level, global behaviors that emerge from this class of models, which we conjecture should apply to crowds more generally.

To extract such global behaviors, we develop a top-down approach by considering the system as a whole and summing the changes in the individual agent dissatisfactions $\Delta H$ to obtain a net global population dissatisfaction functional $H[n(\mathbf{x})]$ (Methods). Integrating over $dn$ and area element $dA$ yields

$$H[n(\mathbf{x})] \equiv F[n(\mathbf{x})] + \int V(\mathbf{x})n(\mathbf{x})dA, \quad (1)$$

where the net frustration effect at location $\mathbf{x}$ is described by $f(n) = \int f'(n)\,dn$, and a local density approximation[24,25] $F[n(\mathbf{x})] \equiv \int f(n(\mathbf{x}))\,dA$ is in this case sufficient for capturing the crowd behavior. This global functional $H[n(\mathbf{x})]$ and the model described above then lead mathematically to the prediction (Methods) that the probability for observing a crowd arrangement with density $n(\mathbf{x})$ will be given by the probability density functional

$$P[n(\mathbf{x})] = Z^{-1}\exp(-H[n(\mathbf{x})]), \quad (2)$$

where $Z$ is an overall normalization constant. Since we cannot measure the function $n(\mathbf{x})$ directly in experimental crowds, we instead consider discrete counts of individuals within equal area bins (quadrats)[26]. Thus, to make contact with experiments we discretize Eq. 1 as $H = \sum_b \left(f_{N_b} + v_b N_b\right)$, where $v_b$ is the average value of the vexation $V(\mathbf{x})$ over bin $b$, and $f_{N_b} \equiv f(N_b/A)A$ approximates the total frustration contribution of bin $b$ of area $A$ (Methods). Substituting this discretization into Eq. 2, the overall probability factors into independent distributions for each bin of the form

$$P_b(N) = z_b^{-1} \frac{1}{N!}(e^{-v_b})^N e^{-f_N}, \quad (3)$$

where $z_b$ is a bin-dependent normalization constant and $N!$ accounts for equivalent configurations among the bins (Methods). Thus, we predict that the fluctuations of the bin counts will be statistically independent and follow a modified Poisson form for each bin. This formulation dramatically reduces the complexity of the system description from tracking each individual to tracking the local density in each bin. Additionally, instead of rules with potentially complex interactions for each agent, the global system behavior of the density is determined by just two functions, $v_b$ and a bin-independent $f_N$. Because this reduction in

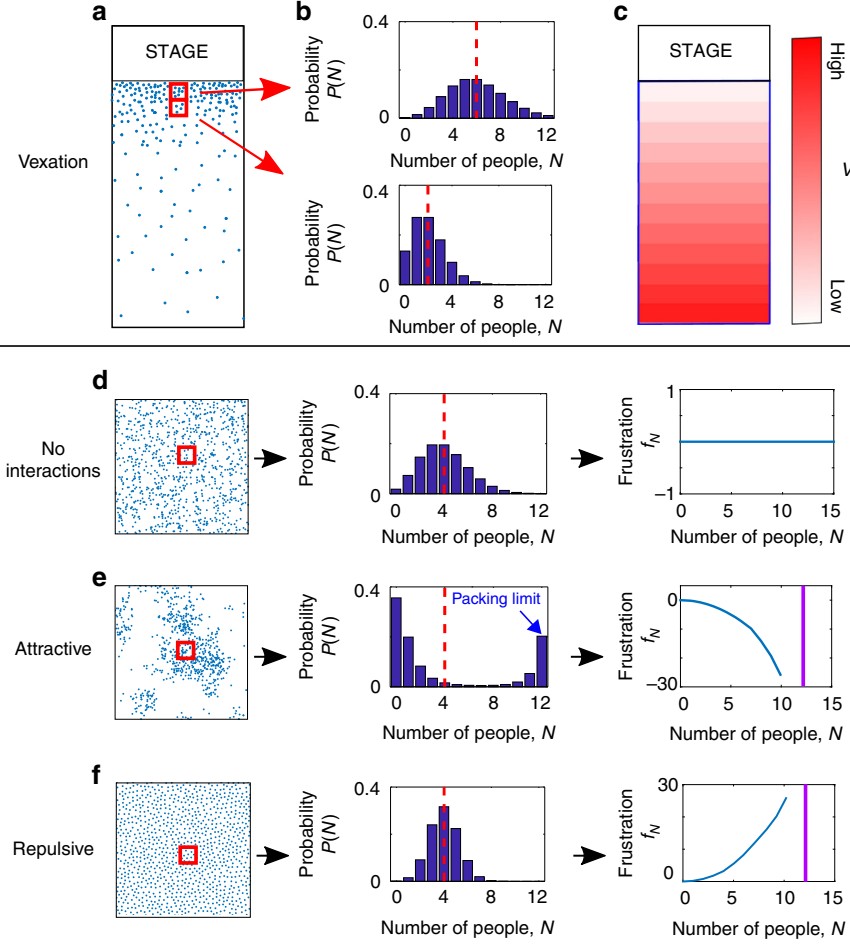

**Fig. 1** Resulting density-functional approach. **a** Schematic of crowd in which agents attempt to get as close to the stage as possible while avoiding overcrowding. **b** In the absence of interactions, the mean of each probability distribution (vertical dashed line) indicates location preference, from which we can extract a bin-dependent vexation functional, $v_b$. **c** Resulting bin-dependent vexations. **d**–**f** Crowds in environments with uniform vexation but with neutral, repulsive, or attractive interactions. For neutral interactions, we expect complete spatial randomness leading to Poisson distributed counts within each bin. The repulsive and attractive interactions are thus reflected in the deviation of the probability distribution from the Poisson form[26]. From these deviations we can extract a bin-independent frustration functional, $f_N$, whose curvature indicates the nature and intensity of the interaction

the number of variables is the result of transitioning to a local-density description as in classical density-functional theory, but now with the modification that interactions are inferred from density fluctuations, we call our approach density-functional fluctuation theory (DFFT).

Remarkably, rather than postulating these functions, they can be extracted directly from measurements of density distributions in each bin. In particular, in the case of neutral interactions ($f_N = 0$), the bin counts will be single-parameter Poisson distributed, as expected for an experiment counting so-called completely spatially random events[26]. From the mean of these distributions one can extract an effective $v_b$ (Fig. 1b, c), or logarithm of the so-called intensity[26], that can arise either from actual preferences for particular locations or from other kinetic interactions with the environment, such as slowing down near barriers[27]. In the case of interactions, such probability distributions can vary substantially from their non-interacting form (Fig. 1d) when the interactions are included (Fig. 1e, f). For example, so-called contagious distributions, which correspond to attractive interactions and show increased variance-to-mean ratios, have been observed[26,28,29]. If the interaction is strongly attractive, groups will form, resulting in a bimodal bin probability distribution corresponding to low and high density regions (Fig. 1e), with the high density region constrained by the packing limit. In contrast,

highly repulsive interactions (Fig. 1f) lead to more uniform distribution of individuals in the crowd[26] and will narrow the bin probability distribution. Finally, from distortions off of the Poisson form, we can determine an effective frustration function $f_N$, without assuming any particular functional form, that describes any local interaction, attractive or repulsive. This formulation holds whether the interaction is directly related to density or to more complex factors such as orientation distributions, as well as higher-order many body interactions (Methods). The power of this approach is that, since $v_b$ is tied to the interactions with the environment and $f_N$ is tied to inter-agent interactions, it may be possible to combine vexations and frustrations from previous measurements to predict future crowd behaviors.

Several conditions must be met when applying this methodology to crowds under realistic circumstances. For example, the system must be sufficiently ergodic. Thus, the time scales for measurements must be longer than the system decorrelation time. In addition, the agent interactions with their environment should be sufficiently independent of the agent density, the agent interactions should be sufficiently independent of location, and both should be stable over the measurement time. Finally, bin sizes must be appropriately chosen. The bins must be large enough to yield reliable estimates of density, as well as to avoid

trivial correlations in neighboring bins, yet small enough that the underlying vexation and local density are nearly constant across each bin.

**Extraction of functionals for model system of walking flies.** To test whether this approach applies to actual populations, we consider a model crowd consisting of wild-type male *Drosophila melanogaster* from an out-bred laboratory stock. It is well know that flies exhibit complex spatial preferences[30,31] and social behaviors[32,33]. Here we seek to determine whether a large crowd of individuals with such complex behaviors indeed can be described within our vexation and frustration framework. The flies are confined in 1.5 mm tall transparent chambers where they can walk freely but cannot fly or climb on top of each other. We record overhead videos of the flies, bin the arena, and use custom Matlab-based tracking algorithms (Methods) to measure the individual bin counts $N_b$ in each video frame. To explore a variety of behaviors, we use arenas of different shapes[30] and apply heat gradients[34] across the arenas to generate different spatial preferences. We find that the flies fully adjust to such changes in their environments after 5 min. We also find that the behavior of the flies changes slowly over a time scale of hours (Methods). We thus take care to make our observations over 10 minute windows during time periods where the behavior is stable.

A top down image of 65 flies in a quasi 1D arena that is uncomfortably heated on the right is shown in Fig. 2a. We find that a bin size of 0.15 cm$^2$, corresponding to the area of approximately 7 flies, ensures that the counts are spatially

independent (Fig. 2b) and that the density does not vary substantially over each bin. We also find that the decorrelation time for $N_b$ is about 5 s (Fig. 2c) indicating the system is sufficiently ergodic over the time scale of our observation windows. We show representative probability distributions $P_b(N)$ for a high and a low density bin in Fig. 2d, e, respectively. We find that the distribution peaks are centered at higher $N$ near the left side of the chamber suggesting lower vexation there. Additionally, the high density probability distribution is significantly narrower than the fitted Poisson distribution, hinting that there are repulsive interactions among the flies.

To validate our description and quantify the vexations and frustrations, we plot what we call as a mnemonic the "pseudo-free energy" $-\ln(N!P_b(N)) = (v_b N + \ln z_b) + f_N$ versus N in Fig. 2f. To determine whether the frustration $f_N$ is indeed universal, we subtract a linear term corresponding to a bin-dependent vexation and normalization constant, $v_b N + \ln z_b$, from each curve. Remarkably, the resulting curves can be made to collapse, indicating that a single, universal frustration function $f_N$ applies equally well to all bins (Fig. 2g). The positive curvature indicates that higher densities are less preferable than expected from non-interacting populations, and thus indicates repulsive interactions. We also show the bin-dependent vexation values $v_b$ used to collapse the curves in Fig. 2h. Finally, as an indicator of the strength of the collapse, we find that modifying the best least-squared fit Poisson distributions by including just eight universal frustration values ($f_0$ through $f_7$) decreases our reduced $\chi^2$ value for 166 degrees of freedom from 8.1 to 0.95. Additionally, our DFFT model is favored by the likelihood ratio test with probability

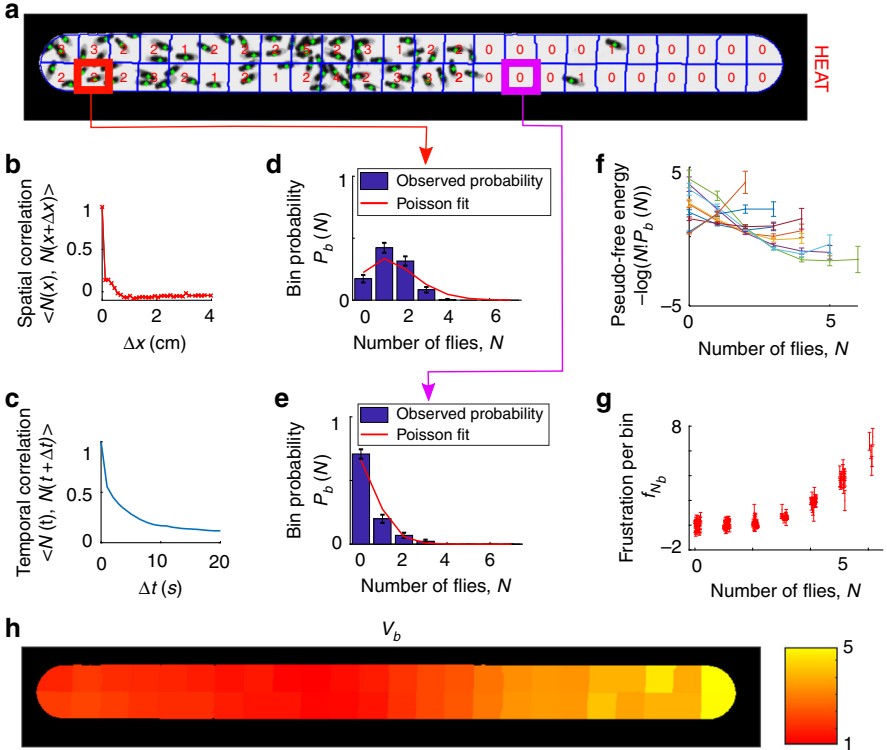

**Fig. 2** Statistical analysis and extraction of functionals for walking fruit fly experiments. **a** Single frame of 65 flies walking in a quasi 1D chamber of dimensions 10 cm × 0.8 cm divided into 48 bins with approximate area 0.15 cm$^2$. Heat is applied on the right side of the chamber so that the temperature varies from 35 °C on the left to 50 °C on the right. **b** Averaged spatial correlation function. **c** Averaged temporal correlation function. **d–e** Probability distributions of the number of flies in the two bins outlined in **a** in red and magenta, respectively. **f** The "pseudo-free energy," $-\ln(N!P_b(N))$, for eight representative bins. The observed positive curvature indicates deviations from the Poisson form and repulsive interactions. **g** Frustration functional, $f_N$, obtained from collapse of the pseudo-free energies for all 48 bins upon removal of the Poisson contributions. **h** Vexation for each bin as measured from the Poisson contributions to the pseudo-free energies. S.d. error bars in **d–f** computed from Bayesian posterior distribution assuming a Dirichlet prior. S.d. errors bars in **g** computed from linear propagation of errors displayed in **f**

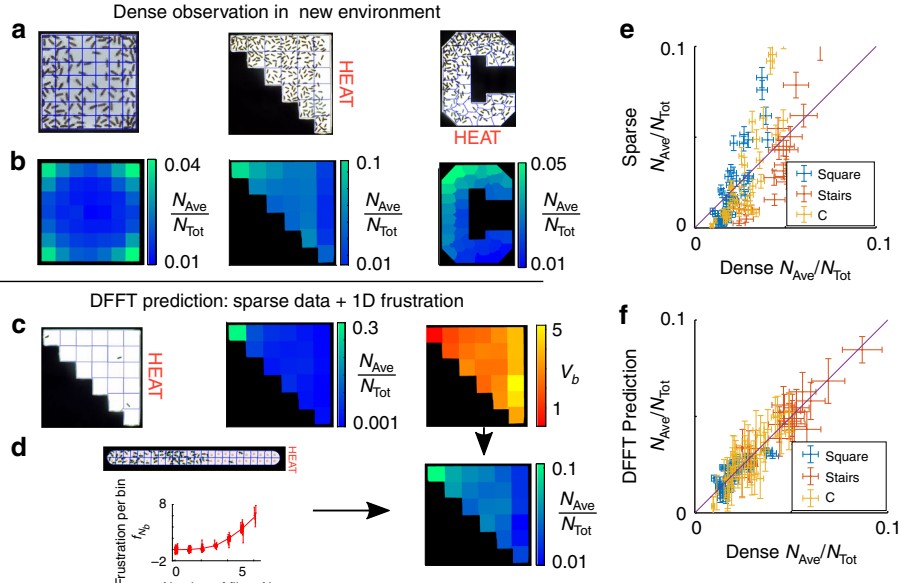

**Fig. 3** Predictions of large crowd distributions in three new environments. **a** Experimental observations of dense crowds (124, 219, and 189 flies) in three chambers with different geometries, two with applications of heat creating temperature differences of up to 20 °C. **b** Measured single-fly probability distributions, $N_{Ave}/N_{Tot}$. **c, d** DFFT protocol applied to the stair-case geometry. **c** Measurement of the density for 3 flies is used to determine the vexation, $v_b$. **d** Combining this vexation with the extracted quasi 1D frustration from Fig. 2 leads to the high density DFFT prediction. **e** Comparison of single-fly probabilities for the sparse and dense populations shows significant population shifts as indicated by a correlation coefficient $r = 0.73$ and a $\sigma_{mean} = 3.8$. **f** DFFT analysis that incorporates interactions predicts the measured dense population distribution within statistical uncertainty ($r = 0.96$ with a $\sigma_{mean} = 1.0$). Vertical error bars correspond to s.d. of bin-occupation distributions and horizontal error bars correspond to s.e.m. of the observed density within a given bin

$p < 0.001$ for accepting the hypothesis that the frustration values should be taken to be zero and a vexation-only model be used. This latter test confirms that the aforementioned reduction in $\chi^2$ is not a result of overfitting (Methods).

**Predictions of crowd density under new circumstances**. An important consequence of the physical independence of $f_N$ from $v_b$ is that it should be possible to use the frustrations extracted from the quasi-1D chamber to predict fly distributions in distinct vexations (Fig. 3). We demonstrate this capability by predicting the measured density distributions for large numbers of flies (on the order of 100) in three distinct geometries and temperature gradients (Fig. 3a). Using measurements of just a few flies in each chamber, we extract density distributions and determine the corresponding vexation $v_b$. Combining this few-fly vexation for each environment with the many-fly frustration $f_N$ extracted from the quasi 1D geometry, we predict the fly distributions under dense conditions. Fig. 3b shows this procedure for the stair-case geometry. We find that the individual fly probability distributions (density normalized by total number of flies) for low and high densities are significantly different (Fig. 3c). In contrast, including the interactions through our DFFT approach predicts a more homogeneous population that matches the observed distribution (Fig. 3d). These results demonstrate that, using our DFFT analysis, it is indeed possible to make accurate predictions by combining vexations from low-density experiments in different environments with a frustration that corresponds to a particular behavior ("mood").

**Frustration used to quantify the "mood" of a crowd**. Conversely, by keeping the environmental conditions fixed and analyzing different time points in the experiments or changing the ratio of male to female flies, the resulting change in "mood" can be quantified by extracting the corresponding functionals. For

example, after spending about six hours in the chamber without food or water, the flies exhibit transient groups or clusters of about 10-20 individuals. This change in behavior is quantified by the different curvatures for the frustrations $f_N$ characterizing the initial (blue curve) and deprived states (red curve) in Fig. 4. The nearly flat frustration associated with this behavior indicates that male flies are willing to surmount their natural repulsion and form higher density groups under deprivation conditions, a previously undocumented spontaneous self-organized change in collective behavior[31,32,35]. Attraction between individuals can be induced by introducing female flies. For groups of flies with equal numbers of males and females which have been separated for several days, we find pair formation (yellow ellipses). This behavior is characterized by a sharp downward curvature in the frustration at low $N$ (yellow curve). Exposing this population to similar deprivation conditions drives formation of larger groups (purple circle) at the expense of pair formation. This behavior is captured by the shift of downward curvature in the frustration to larger bin occupations of $N \approx 7$ (purple curve). These data establish that the DFFT approach has the power to detect and quantify changes in social behaviors.

## Discussion

Collectively, these results demonstrate that top-down approaches are a promising method for predicting crowd distributions and quantifying crowd behaviors. The DFFT analysis that we present is particularly powerful because it separates the influence of the environment on agents from interactions among those agents. This separation then enables predictions of crowd distributions in new situations through mixing and matching of the vexations and frustrations from previous observations in different scenarios. In addition, the real-time quantification of frustrations opens the door to tracking behavioral changes and potentially extrapolating the time evolution of frustrations to anticipate future behaviors.

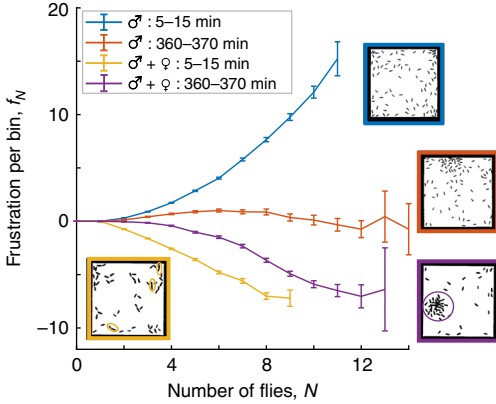

**Fig. 4** Extracting frustrations to quantify changing behavior. Frustrations measured for flies in a 4 cm square chamber. The experiment duration was seven hours. The frustrations were extracted from two different 10 minute intervals corresponding to the initial and final stages of experiments on two different populations. The blue curve (90♂) exhibits a positive curvature at all occupancies, indicating an aversion to crowding at all densities. The red curve characterizes interactions for the same population 6 hours later. The lower curvature indicates significantly reduced aversion to grouping. The yellow curve (30♂ + 25♀) exhibits a downward curvature at low occupations, reflecting mating interactions between pairs of flies (yellow ellipses). At higher occupancies, the lack of curvature indicates a more neutral response to changes in occupation number. Finally, the purple curve characterizes interactions for the same mixed-sex population 6 hours later. The downward curvature shifts to higher occupancies and is followed by a region of positive curvature. The corresponding inflection point indicates a preference for group formation with a density of about eight flies per bin. S. d. error bars calculated from the maximum likelihood (ML) covariance matrix of DFFT distribution in Eq. 3

There are a number of directions in which the formal framework suggested here can be extended, paralleling developments from the traditional density-functional theory literature. Extensions to time-dependent DFT methods (TDDFT)[36,37] would enable the prediction of situations in which crowds gather and disperse in response to changes in the environment. This approach would also apply to situations in which the center of mass of the entire group is moving as whole, such as in herd migration and bacterial and insect swarming. Moreover, by including the local current density ("flow") in the functional, such approaches may even be able to describe crowds where correlated subgroups move with different local velocities, such as in flocks of birds. Likewise, extensions to multicomponent DFT[38] would enable corresponding predictions and observations in crowds composed of distinct groups exhibiting interactions such as inter-group conflict, predator-prey relations, or mating behavior.

Should these results extend to human populations, the implications are profound. From publicly available video data of people milling in public spaces, this approach could predict how people would distribute themselves under extreme crowding. Additionally, a simple application running on a hand-held device could easily measure density fluctuations and extract functionals that are indicative of the current behavioral state or mood of the crowd. Through comparison with a library of functionals measured from past events, such an application could provide early warning as a crowd evolves towards a dangerous behavior. Finally, given the recent proliferation of newly available cell-phone and census data[39,40] these approaches may also extend to population flows on larger scales, such as migration. Here, vexations could correspond to political or environmental drivers and frustrations to population pressures. The resulting

predictions of migration during acute events would enable better planning by all levels of government officials, from local municipalities to international bodies[40,41], with the potential to save millions of human lives.

## Methods

**Global dissatisfaction functional $H[n(\mathbf{x})]$.** The main text describes a net global population dissatisfaction functional $H[n(\mathbf{x})]$. To derive this functional, we begin by considering a deterministic model, in which agents reject or accept potential moves with unit probability according to whether $\Delta H \equiv (V(\mathbf{x}') + f'(n(\mathbf{x}'))) - (V(\mathbf{x}) + f'(n(\mathbf{x})))$ is positive or negative, respectively. In such a model, it is clear that equilibrium is attained and all motion ceases when $\Delta H = 0$ for all pairs of points $\mathbf{x}$ and $\mathbf{x}'$. This statement is equivalent to the combination $V(\mathbf{x}) + f'(n(\mathbf{x}))$ attaining some constant value $\mu$ across the system,

$$V(\mathbf{x}) + f'(n(\mathbf{x})) = \mu. \tag{4}$$

This equation corresponds precisely to the Lagrange-multiplier equation for minimization of the functional

$$H[n(\mathbf{x})] \equiv \int f(n(\mathbf{x}))\, dA + \int V(\mathbf{x}) n(\mathbf{x})\, dA, \tag{5}$$

subject to the constraint of fixed number of agents $N = \int n(\mathbf{x})\, dA$, with $\mu$ being the corresponding Lagrange-multiplier. Here, $\mu$ plays an analogous role to the "chemical potential" from Statistical Physics.

**Probability density functional $P[n(\mathbf{x})]$.** To make the transition to the probability functional $P[n(\mathbf{x})]$, we note that the stochastic model described in the text maps directly onto a particular Markov chain. Each step on this chain corresponds to a three-stage process. First, (a) an agent is selected at random to consider a possible move from current location $\mathbf{x}$. Selecting a random agent at each time step allows agents to adjust their locations at equal rates. In this approach, choosing the physical time interval between Markov steps to be inversely related to the number of agents preserves the time scale of the overall crowd dynamics. Second, (b) a location $\mathbf{x}'$ nearby $\mathbf{x}$ is selected at random as a move to be considered by the given agent. We note that for this work, we assume that the new location $\mathbf{x}'$ is selected in a symmetric way so that that agents at $\mathbf{x}$ contemplate moves to $\mathbf{x}'$ with the same probability that agents at $\mathbf{x}'$ contemplate moves to $\mathbf{x}$. This assumption seems most plausible given the systems we consider here. Other selection criteria, however, are possible and would modify the distribution below. Finally, (c) the contemplated move is accepted or rejected according to the probability $1/(e^{\Delta H} + 1)$, where $\Delta H$ is defined specifically as the change in the value of the functional described in Eq. 5 as a result of the move.

There are two critical things to note about this Markov chain. The first is that it gives a very natural description of agent behavior. The second is that it corresponds precisely to the standard Metropolis-Barker algorithm[42,43] for drawing random samples from the Boltzmann distribution $P \propto \exp(-H)$ for a Hamiltonian $H$. Thus, under our proposed motion model, the population itself naturally samples from the distribution quoted in the text,

$$P[n(\mathbf{x})] = Z^{-1} \exp(-H[n(\mathbf{x})]). \tag{6}$$

**Discretization $H = \Sigma_b (f_{N_b} + v_b N_b)$.** To arrive at the discretization described in the text, it is important to note that the density $n(\mathbf{x})$ appearing in the probability functional $P[n(\mathbf{x})]$ corresponds to the fluctuating crowd density, as opposed to the average density $n_{\text{ave}}(\mathbf{x})$. As such, in practice, this density must be described in terms of the discrete locations $\mathbf{x}_a$ of all agents $a$ in the crowd at any give time. The most natural description for the associated density operator is

$$n(\mathbf{x}) = \sum_a \delta^{(\sigma)}(\mathbf{x}, \mathbf{x}_a), \tag{7}$$

where $\delta^{(\sigma)}(\mathbf{x}, \mathbf{x}_a)$ is a function describing the range over which the presence of an agent at $\mathbf{x}_a$ contributes to the density $n(\mathbf{x})$ at point $\mathbf{x}$. To conserve number of agents, this function must integrate to unity. The analysis carried out in the text divides space into bins $b$ of area $A_b$, and estimates the density in each bin as $n = N_b/A_b$ where $N_b$ corresponds to the total number of agents in bin $b$. This definition sets the range function as

$$\delta^{(\sigma)}(\mathbf{x}, \mathbf{x}_a) \equiv \begin{cases} \frac{1}{A_b} & \text{if } \mathbf{x} \text{ and } \mathbf{x}_a \text{ are in the same bin } b \\ 0 & \text{otherwise} \end{cases} \tag{8}$$

To capture relevant variations in vexation and density, the bins cannot be selected so large that these quantities vary significantly across each bin. Alternately, to avoid missing the effects of nearby agents, the bins cannot be selected to be smaller than the agent's interaction range.

Finally, combining equations 5, 7, and 8, yields

$$H[n(\mathbf{x})] = \sum_b f_{N_b} + \sum_b \nu_b N_b, \qquad (9)$$

where $f_{N_b} \equiv f(N_b/A_b)A_b$ and $\nu_b \equiv \int_b V(\mathbf{x})\,dA/A_b$.

**Bin occupation probability distributions $P_b(N)$.** To arrive at the final discrete probability expression in the text, there are now two routes. One can directly insert Eq. 9 above into Eq. 2 from the main text, or one can employ Eq. 9 directly to compute $\Delta H$ to determine the probabilities for moves. In the latter case, the predicted probability distribution becomes exact so long as we interpret $f'(n)$ in the main text at points $\mathbf{x}'$ and $\mathbf{x}$ to represent forward and reverse finite difference derivatives $f'_+(n(\mathbf{x}')) = (f(n(\mathbf{x}') + \Delta) - f(n(\mathbf{x}')))/\Delta$ and $f'_-(n(\mathbf{x})) = (f(n(\mathbf{x})) - f(n(\mathbf{x}) - \Delta))/\Delta$, respectively, where $\Delta \equiv 1/A_b$. Finally, because the Boltzmann factor above gives probabilities for individual arrangements of agents among bins, we must account for the multiple ways to realize a set of bin counts $\{N_b\}$ by permuting individuals among the bins. Multiplying by the combinatorial factor $N_{\text{tot}}!/(N_1!\ldots N_b!\ldots)$, we find

$$P(\{N_b\}) = \frac{N_{\text{tot}}!}{Z} \prod_b \frac{e^{-f_{N_b} - \nu_b N_b}}{N_b!}, \qquad (10)$$

where $Z$ is a normalization factor.

As described in the text, we note that the form of the joint probability distribution above predicts the occupations of different bins to be very nearly statistically independent. The only deviation from complete statistical independence comes from the constraint of a fixed total number of agents $N_{\text{tot}} = \sum_b N_b$. Due to this constraint, the probability distribution is difficult to use in making predictions. We can overcome this difficulty using a standard technique from statistical physics. Specifically, introducing a factor $e^{\mu N_{\text{tot}}}$ removes the constraint without significantly affecting the calculated local distributions. As a result, the individual bin distributions then become statistically independent and of the form

$$P_b(N) = z_b^{-1} \frac{1}{N!} \left( e^{-(\nu_b - \mu)} \right)^N e^{-f_N}. \qquad (11)$$

In statistical physics this mathematical transformation corresponds to using a Grand Canonical Ensemble[44] to simplify statistical calculations. Physically, this approach corresponds to relaxing the constraint of a fixed number of agents by allowing exchanges between the system being considered and a large reservoir whose vexation is controlled by $\mu$. Mathematically, we can add and subtract a constant within the exponent, $(\nu_b - c - (\mu - c))$ without affecting the distribution. Accordingly, we redefine $\nu_b$ and $\mu$ with a constant shift such that $\nu_b \leftarrow \nu_b - c$ and $\mu \leftarrow \mu - c$ and, further, choose $c$ so that $\mu = 0$, resulting in Eq. 3 in the text. Note that motion between bins is controlled only by differences in vexations, so that none of this affects the dynamics represented in our analysis. When considering a different number of agents in the same chamber, however, $\mu$ will take on a different value and so $\mu - c$ can no longer be set to zero. Accordingly, to predict distributions for new numbers of flies, we employ Eq. 11 above and adjust $\mu$ so that the vexation of the associated reservoir fixes the new total number of flies.

**Orientation and higher-order many-body interactions.** Remarkably, our conclusions hold also for plausible models in which the inter-agent interactions are not explicitly expressed in terms of the local density $n(\mathbf{x})$. To see this, we can consider the same behavioral rule of moves accepted according to probability $1/(e^{\Delta H} + 1)$, but with $H$ now defined as a sum of two parts,

$$H \equiv U(\mathbf{x}_a) + \sum_a V(\mathbf{x}_a), \qquad (12)$$

where $V(\mathbf{x})$ is the usual vexation function for the individual agents, and now $U(\mathbf{x}_a)$ is some potentially complex many-body interaction of finite range depending explicitly on the locations of all of the agents $\mathbf{x}_a$.

As above, the form of the Markov chain associated with the move model leads directly to the Boltzmann distribution $P(\mathbf{x}_a) = Z^{-1}e^{-H}$. To recover the frustration-vexation probability form analyzed throughout the text, we now follow the standard Statistical Mechanics approach of defining an pseudo-free-energy functional by integrating out internal degrees of freedom. Specifically, we will keep the bin occupancies constant while integrating over all arrangements of agents consistent with these occupancies. For sufficiently small bins in which vexation does not vary significantly, we again find to a good approximation $\sum_a V(\mathbf{x}_a) = \sum_b \nu_b N_b$, so that vexation simply gives a constant factor. Next, for sufficiently large bins, the net contributions to $U(\mathbf{x}_a)$ from interactions occurring within the bins will be large compared to the boundary effects from contributions from interactions crossing bin boundaries. Thus, we can imagine decomposing the overall interaction into a sum over the bins of the interactions just among agents $a$ within each bin $b$, $U(\mathbf{x}_a) = \sum_b U(\{\mathbf{x}_a\}_{a \in b})$, where we can improve accuracy by repeating the same agent locations $\{\mathbf{x}_a\}_{a \in b}$ in neighboring bins (so-called periodic boundary conditions).

Combining these approximations, and summing over all ways to assign agents to bins with counts $\{N_b\}$ and over all possible locations for the agents within each bin, yields the same frustration-vexation form considered throughout the text,

$$P(\{N_b\}) = Z^{-1} \binom{N}{N_1! \ldots N_B!} \left( \prod_b e^{-f_{N_b}} \right) e^{-\sum_b \nu_b N_b}, \qquad (13)$$

where $B$ is the total number of bins, and

$$e^{-f_N} \equiv \int_A \ldots \int_A e^{-U(\mathbf{x}_1, \ldots \mathbf{x}_N)} \, dA_1 \ldots dA_N \qquad (14)$$

defines the effective bin-frustration functional $f_N$ as an $N$-dimensional integral over the area of a single bin (with periodic boundary conditions applied to the interactions). Finally, we note that the above generalizes naturally to orientation-dependent interactions by considering the coordinates $\{\mathbf{x}_a\}$ to include orientation, as well as spatial coordinates. If the vexation is orientation-independent, we recover precisely the form above. Otherwise, the entire framework generalizes naturally to consideration of joint location-orientation densities $n(\mathbf{x}, \theta)$.

**Experimental setup.** All experiments were performed 3–15 days post-eclosion using common fruit flies (*D. melanogaster*) from an out-bred laboratory stock reared at room temperature on a 12 h/12h day-night cycle. Flies are anesthetized using $CO_2$ and sorted within a few days post-eclosure. We wait for 24 h after sorting before running experiments. Most observations started between 1–5 h after the light was turned on. The experiment chambers are constructed by sandwiching a 1.5 mm thick aluminum frame between two transparent acrylic sheets. The chamber is suspended above an LED light table. Holes in the upper acrylic sheet allow for the introducing flies via aspiration from above. To heat the chambers, $2\,\Omega$ high-power resistors are adhered using JB Weld to the aluminum sheet and powered by a variable power supply. On the opposite side of the sheet, a beaker of ice water is used as a heat sink. Chamber temperature is measured for two locations using a contact thermometer to ensure no more than 2 degrees Celsius drift and consistent temperature gradients between trials. We heat one side of the chamber to temperatures between 40–50 degrees Celsius[34]. The opposing side of the chamber is connected to a heat sink and kept at temperatures between 25–35 degrees Celsius. We find that the resulting temperature gradient drives a strong avoidance behavior for the hotter wall while avoiding fly death as the flies avoid the high-temperature region. A video camera (AVT Marlin, Andover, MA) records overhead images of flies at frame rates around 30 fps and relays these images to a computer where they are analyzed by a custom MATLAB program in real-time. The entire apparatus was enclosed in a black box to prevent biases introduced by ambient light or additional visual cues.

**Image analysis.** To label fly centroids, images were thresholded to find fly silhouettes. For high density experiments, large groups become common and a more sophisticated approach is necessary to separate clusters, which may be as large at 10 flies. First, the images of several individual flies are combined to make a single, averaged fly mask. This mask is then convolved with images of fly groups. The best fits for these convolutions are used to approximate the locations of flies whose silhouettes overlap. (For additional details, see code provided under Code Availability statement below.) Labeling is then manually checked and we find this technique robust enough to label male flies with 0.25 % error or 1 in 400 flies mislabeled. The mating flies required extensive manual corrections due to changes in the fly postures and the polydispersity of fly sizes, since females are larger than males. For the analysis in this paper we sampled these positions at intervals of 1 s.

Due to wall-exclusion effects, the area of a chamber is different from the area accessible by the centroid of a fly. We thus exclude the outer area of the chamber that corresponds to approximately half the width of a fly. Areas of the bins are then extracted using images from the experiment.

To demonstrate another method for tracking flies that only measures local densities, a simpler method was used for counting flies in the "C" shaped chamber. After thresholding, the number of pixels corresponding to a fly were summed in each bin and then a discrete fly density was assigned to each bin using knowledge of the total number of flies in the chamber. This method has the advantage of computational speed, but weights larger flies more heavily and requires reanalysis for different bin sizes.

**Measurement timing and thermal ramp protocol.** Observations for Fig. 3 were conducted using time intervals from approximately 5–15 min after being introduced into the chamber so that the flies could explore their new chamber and adjust to a steady state. To measure the vexation of the square experiment, we performed 12 separate single fly measurements each lasting 10 min. Similar results are obtained after three flies are used over a single 10 min period. Thus, measurements of vexation in the "C" and stair shaped chambers used two and three concurrent flies and only needed a single ten minute observation to measure the vexation.

To probe the changing fly behaviors shown in Fig. 4, we track the flies for up to 9 h before flies begin to die from deprivation[45,46]. To test whether fly behavior is changing over our standard 10 minute time windows, we compare the probabilities, $P_b(N)$, from the first 5 min of the window with the last 5 min and find that they are

consistent. The only exception to this is during the very first 5 min after the flies are introduced into the chamber as they become oriented to their new environment that we do not include in our analysis. To elicit different behaviors and location preferences with the same population of flies, we apply a heat gradient to generate an avoidance behavior[34] starting at 20 min after being introduced to the chamber. By minute 30, the chamber has reached a steady temperature and we observe that the flies exhibit an approximately constant average distribution. At minute 40, we turn the heat off and let it adjust to room temperature for the remainder of the experiment. Throughout these observations, we qualitatively observe several different behaviors. For the first 5 min, flies are most active and their frustration has a slightly higher positive curvature than the frustration for the 5–15 min period. When the chamber is heated, the frustration stays approximately the same despite the drastic change in the vexation. After the chamber cools down, flies enter a readjustment phase where they are much less active. After this readjustment phase, however, flies again exhibit behavior similar to that from the 5–15 min interval. By 6 h, flies in all the experiments switch to a grouping behavior as shown in Fig. 4.

**Validation of assumptions underlying theoretical analysis.** As mentioned above, we made some general assumptions developing our theory which we now validate for the walking fly system. First, to verify attainment of equilibrium and sufficient ergodicity, we consider the normalized autocorrelation function

$$c_{\mathrm{T}}(\Delta t) \equiv \frac{\left\langle \sum_b N_b(t) N_b(t+\Delta t) \right\rangle_t}{\left\langle \sum_b N_b(t) N_b(t) \right\rangle_t}, \tag{15}$$

where $\langle \ldots \rangle_t$ indicates average over all times. This function shows the expected rapid exponential decay (Fig. 2c), and has an integral which gives the decorrelation time $\tau = 0.92\,s$. Indeed, we find this time to be quite short, typically on the order of a few seconds, for all of our experimental runs. This decay time is two orders of magnitude faster than the typical run time and does not vary significantly when computed in different time sub-windows, strongly suggesting rapid mixing and stationarity of the random process, thereby allowing the interchange of time and ensemble averages, and establishing the existence of equilibrium in the timescales under study. Our videos thus represent hundreds of independent samples drawn from the equilibrium ensemble underlying our analysis.

We next consider whether the bins are truly independently distributed as expected in Eq. 3. Accordingly, we consider the normalized time-averaged spatial-correlation function

$$c_{\mathrm{S}}(\Delta) \equiv \frac{\left\langle \sum_b N_b(t) N_{b+\Delta}(t) \right\rangle_{b,t}}{\left\langle N_b(t) N_b(t) \right\rangle_{b,t}}, \tag{16}$$

where $\langle \ldots \rangle_{b,t}$ indicates average all times and bins, and $\Delta$ is the two-dimensional vector displacement between bins (Fig. 2b)). The data show essentially no correlation between bins, thereby verifying the product form of the global bin distribution function in Eq. 3 in the main text. This confirms not only that we have chosen appropriately sized bins but also, more fundamentally, establishes that there are little or no fly–fly interaction effects between bins, so that the local density approximation (LDA) form for the frustration, $F[n(\mathbf{x})] = \int f(n(\mathbf{x}))dA$, indeed gives a good representation of the behavior of the fly populations at scales greater than $0.15\,cm^2$.

**Parameter estimation.** To estimate the frustration and vexation for the crowds in our experiments, we start by constructing the posterior function $P(f_N, v_b | N_b(t))$, which represents the relative likelihood of different parameter choices for our model given the data (number counts within each bin) that has actually been observed. Then, to find the a posteriori estimate of the parameters, we maximize this likelihood by performing a numerical gradient minimization of

$$-\ln P(f_N, v_b | N_b(t)) = C + TB\Bigg( \langle \ln z_b \rangle_b + \Big\langle v_b N_b(t) + \ln N_b(t)! + f_{N_b(t)} \Big\rangle_{b,t} \Bigg) + \sum_N \frac{f_N^2}{2\sigma^2} + \sum_b \frac{v_b^2}{2\sigma^2}, \tag{17}$$

where $C$ is an irrelevant normalization constant, $B$ corresponds to the total number of bins in the system, $T$ the total number of independent time samples employed, and $\langle \ldots \rangle_b$ and $\langle \ldots \rangle_{b,t}$ represent averages over either all bins or bins and times, respectively. Finally, for the last two terms, $\sigma$ accounts for the range about zero of a Gaussian prior distribution on the frustration and vexation parameters. This Gaussian prior distribution reflects the fact that the frustration and vexation parameters $v_b$ and $f_N$ can in principle take any real value, but in practice generally fall in a range on the order of from about $-15$ to 15 because these parameters enter as exponentials in our probability models. Because the amount of data that we handle is on the order of tens of thousands of frames, the likelihood peaks strongly around its maximum, and the precise form of the Gaussian prior is largely irrelevant. Indeed, changing the value of $\sigma$ from a reasonable value of 15 to an unreasonably small value of 1, only changes our final results for the frustration by 11.4%. Throughout the rest of our work, we take $\sigma = 15$.

**Uncertainty in parameter estimation.** The sharp peaks associated with the large amount of data ensure the accuracy of the asymptotic Gaussian approximation, in which the joint probability distribution representing the range of parameters supported by the data is a multivariate Gaussian distribution. As a result, the associated covariance matrix of uncertainties in the parameters is the inverse of the Fisher information matrix $I$ (i.e., the second derivative of $-\ln P$ evaluated at the location of its maximum). The matrices of parameter uncertainties and cross-correlations among them are computed as follows. For our full DFFT model, with vexation and frustration, and the simple Poisson model, with vexation only, we calculate the inverses of the following matrices, respectively,

$$I_{\mathrm{DFFT}}(\{f_N\}, \{v_b\}) = \begin{pmatrix} \left[ I_{ff} \right]_{N_{\max} \times N_{\max}} & \left[ I_{fv} \right]_{N_{\max} \times B} \\ \left[ I_{fv}^T \right]_{N_{\max} \times B} & \left[ I_{vv} \right]_{B \times B} \end{pmatrix}, \tag{18}$$

and

$$I_{\mathrm{Poisson}}(\{v_b\}) = \left[ I_{vv} \right]_{B \times B}, \tag{19}$$

where the matrix elements of each block are

$$\left[ I_{ff} \right]_{N,N'} = T\delta_{NN'}\left( \sum_{\tilde{b}} P_{\tilde{b}}(N) - \sum_{\tilde{b}} \left( P_{\tilde{b}}(N)P_{\tilde{b}}(N') \right) \right) \tag{20}$$

$$\left[ I_{fv} \right]_{N,b} = T P_b(N)\left( N - \sum_{\tilde{N}} \tilde{N} P_b(\tilde{N}) \right) \tag{21}$$

$$\left[ I_{vv} \right]_{b,b'} = T\delta_{bb'}\left( \sum_{\tilde{N}} \tilde{N}^2 P_b(\tilde{N}) - \left( \sum_{\tilde{N}} \tilde{N} P_b(\tilde{N}) \right)^2 \right). \tag{22}$$

Here, $P_b(N)$ is defined as in Eq. 3 in the main text, $T$ again represents the total number of independent time frames, and the " $\sim$ " indicates internal summation indices.

Finally, a subtle, but important, ambiguity arises in the extraction of frustrations and vexations. Specifically, because the exponent in the observed probabilities for each bin takes the form $(\ln z_b + v_b N + f_N)$, making the replacements $(v_b \to v_b - \alpha;\ z_b \to z_b - \beta;\ f_N \to f_N + \beta + \alpha N;)$ leaves the predictions of the model unchanged, and any choice of parameters corresponding to these replacements represents the data equally well. As a result, the Fisher matrices described above are singular. To resolve this "gauge invariance" and remove the singularity, we must break the symmetry among equivalent models by adding two constraints (one for $\alpha$ and one for $\beta$) to our choice of $f_N$. Here, we do this by enforcing the natural choice that $f_0 \equiv 0$ and $f_1 \equiv 0$, corresponding to the convention that that the frustration does not affect the probability for bins with either $N = 0$ or $N = 1$ flies. Finally, in terms of the information matrices above, implementing this constraint corresponds to dropping the first two rows and columns associated with these parameters from the $I_{\mathrm{DFFT}}$ matrix.

**Uncertainty in predictions of average occupations.** With the uncertainties in the extraction of the vexation and frustration parameters from above, we next determined the uncertainties in our predictions of the average bin occupations for large populations in new arenas. The predicted mean densities are

$$\bar{N}_b = \sum_{N=0}^{N_{\max}} N P_b(N) = \frac{1}{z_b}\sum_{N=0}^{N_{\max}} N \frac{e^{-(v_b-\mu)N - f_N}}{N!}, \tag{23}$$

where the normalization is

$$z_b = \sum_{N=0}^{N_{\max}} \frac{e^{-(v_b-\mu)N - f_N}}{N!}, \tag{24}$$

where $P_b(N)$ is the probability of having $N$ flies in bin $b$, $v_b$ is the vexation in bin $b$, and $f_N$ is the frustration associated with having $N$ flies in a bin. We accordingly computed the associated uncertainties using standard linearized error propagation as

$$\sigma(\bar{N}_b) = \sqrt{\left( \frac{\partial \bar{N}_b}{\partial v_b} \right)^2 \mathrm{var}(v_b) + \sum_{N,N'=2}^{N_{\max}} \frac{\partial \bar{N}_b}{\partial f_N}\frac{\partial \bar{N}_b}{\partial f_{N'}}\mathrm{covar}(f_N, f_{N'})}, \tag{25}$$

where $\mathrm{var}(X)$ and $\mathrm{covar}(X, Y)$ represent the variance of random variable $X$ and covariance between $X$ and $Y$, respectively, as determined by the inverse of the Fisher information matrix as discussed above. Finally, the derivatives needed in

Eq. 25 are

$$\frac{\partial \bar{N}_b}{\partial v_b} = -\left(\langle N_b^2 \rangle - \bar{N}_b^2\right), \tag{26}$$

and

$$\frac{\partial \bar{N}_b}{\partial f_N} = -\left(\frac{N_b - \bar{N}_b}{Z}\right)\frac{e^{-N_b(v_b-\mu)-f_{N_b}}}{N_b!}, \tag{27}$$

where $\langle N_b^2 \rangle \equiv \sum_N N^2 P_b(N)$ with $P_b(N)$ as defined above.

A few technical notes are in order to understand the terms present in Eq. 25. First, note that cross-correlations between vexations in different bins are not relevant because $\bar{N}_b$ depends solely on $v_b$ and not on vexations from other bins. Also, cross-correlations between extracted vexations $v_b$ and frustrations $f_N$ are zero in our case because we extract the vexations and frustrations from different, and thus independent, experiments when making our predictions for average occupations. Finally, the uncertainties in $f_0$ and $f_1$ are not included because these uncertainties are zero due to the gauge choice discussed in the section above.

**Uncertainty in experimentally measured bin statistics.** For each independent bin, we obtain from the experiment a sequence of length $N_T$ with elements each corresponding to a bin occupation that can range from zero to the maximum packing of files, $N = 0,\ldots,N_{max}$. From this data, we hope to extract probability parameters $p_N$ describing the bin occupation distributions studied in the main text. For simplicity of notation, we here use lower case $p$ to denote experimentally measured probabilities.

To account for time-correlations in bin occupancies, particularly at high frame rates, we down-sample at intervals given by the decorrelation time $\tau$ and actually consider uncorrelated sequences of length $T = N_T/\tau$. The data then correspond to the result of a random process of making $T$ independent selections among $N_{max} + 1$ possible bin occupations. Thus, for each bin, the probability of observing a given data sequence becomes the multinomial distribution,

$$\binom{T}{h_0 \cdots h_{N_{max}}} p_0^{h_0} \cdots p_{N_{max}}^{h_{N_{max}}}, \tag{28}$$

where $h_N$ represents the number of times ("hits") we observe each of the possible occupancies $N$.

To extract the underlying uncertainties, we note that Bayes' theorem gives the following distribution for the probability parameters to take the values $\{p_N\}$ given the actually observed counts $\{h_N\}$,

$$P(\{p_N\}|\{h_N\}) = \frac{P(\{h_N\}|\{p_N\})P(\{p_N\})}{P(\{h_N\})} \propto \left(\prod_{n=0}^{N_{max}} \frac{p_n^{h_n}}{h_n!}\right) P(\{p_N\}). \tag{29}$$

This posterior probability is proportional to an undetermined prior probability $P(\{p_N\})$ describing our a priori expectations for the values of the $\{p_N\}$ parameters. However, as per our discussion surrounding Eq. 17 above, in the large $T$ limit, the Poisson-like product factor in Eq. 29 above will be highly peaked, and the unknown prior $P(\{p_N\})$ will not have a substantial effect on the posterior distribution.

To completely eliminate the effects of unwarranted assumptions entering through our choice of prior, we assume an uninformative prior distribution that is consistent with the invariance of the probability values under the inclusion of new samples, and choose the multivariate generalization of Haldane's uninformative improper prior distribution[47],

$$P(\{p_N\}) = \frac{1}{\prod_{n=0}^{N_{max}} p_n}. \tag{30}$$

With this choice, upon normalization, Eq. 29 becomes the Dirichlet distribution,

$$P(\{p_N\}|\{h_N\}) = \Gamma\left(\sum_{n'=0}^{N_{max}} h_{N'}\right) \prod_{N=0}^{N_{max}} \frac{p_N^{h_N-1}}{\Gamma(h_N)}, \tag{31}$$

where $\Gamma(x)$ is the Gamma function. This distribution yields expected values for the probabilities equal precisely to the observed frequencies $\bar{p}_N = h_N/T$. The variances of this distribution, then give our desired uncertainties,

$$\sigma(p_N) = \sqrt{\frac{h_N(T - h_N)}{T^2(T+1)}} = \sqrt{\frac{\bar{p}_N(1 - \bar{p}_N)}{T+1}}. \tag{32}$$

Note that when $T$ is large and $\bar{p}_N \ll 1$, the uncertainties correspond to what we would naïvely expect from Poisson counting, namely an uncertainty of $\sqrt{h_N}$ in the counts, corresponding to an uncertainty of $\sqrt{h_N}/T = \sqrt{\bar{p}_N/T}$ in the extracted probabilities. Such an analysis, however, misses the important factor of $\sqrt{1 - \bar{p}_N}$ and leads to significant errors in our case.

Finally, for the uncertainty in the experimental average occupation $\bar{N}_{expt} = \sum_N N p_N$, the corresponding variance is

$$\text{var}\left(\bar{N}_{expt}\right) = \sum_{N \neq N'} NN' \text{covar}(p_N, p_{N'}) + \sum_N N^2 \sigma(p_N)^2, \tag{33}$$

where the needed covariances of the Dirichlet distribution are

$$\text{covar}(p_N, p_{N'}) = \frac{-h_N h_{N'}}{T^2(T+1)} = \frac{-\bar{p}_N \bar{p}_{N'}}{T+1} \tag{34}$$

**Code availability**. Readers can access the code related to parameter estimation and crowd density predictions by going to (https://github.com/MendezV/DFFT) or to (https://doi.org/10.5281/zenodo.1285931). Readers can also access code related to image analysis procedures by visitng (https://github.com/yunuskink/Fitfly-fly-tracking) or (https://doi.org/10.5281/zenodo.1304326). There are no access restrictions to this software.

**Data availability**. The fly density data that support the findings of this study are available in the Open Science Framework database at (https://doi.org/10.17605/OSF.IO/7UBZ2).

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

## Acknowledgements

The authors thank Xiaoning Wang, Marc-Antoine Bouvattier, Alonso Botero, Tom Corwin, Tom Mifflin, Greg Godfrey, and Nathan Sitaraman for their help with the initial stages of the project. We further thank the Cohen and Arias groups for discussions throughout this work. The work was primarily funded by the Army Research Office Army-ARO W911NF-16-1-0433. J.F.M. was also supported in part by an Office of the Vice President for Research at the University of Los Andes. Y.K. was also supported in part by funding from the National Science Foundation Graduate Research Fellowship Award No. DGE-1650441.

## Author contributions

J.F.M.V.: Development and implementation of analyses to extract vexations, frustrations, and predicted mean occupations. Analysis of statistical uncertainties in all of these quantities and also in the extraction of bin-occupancy distributions from experimental data. Final display format for cross-correlating predicted mean occupancies with predictions. Theoretical parts of the Methods Section. Significant contributions to main text. Y.A.K.: Design and implementation of experiments along with development of image analysis techniques. Running data through analyses provided by Méndez Valderrama. Experimental parts of the Methods Section, design, and implementation of figures, and an early draft of the manuscript. Significant contributions to main text. Y.A.K. and J.F.M.V. contributed equally to this work. J.S.: Co-development of underlying Markov chain, proper accounting for degeneracy factor, identification of multinomial and Poisson distributions for the non-interacting case, initiation of use of maximum-likelihood estimation and Bayesian uncertainty techniques. I.C.: Significant input into design of experiments, and primary responsibility for main text. T.A.A.: Development of underlying motion model, density-functional theory analysis, and prediction of form of population fluctuations. Co-development of underlying Markov chain. Complete early draft of manuscript, and significant contributions to main text.

## Additional information

**Competing interests:** The authors declare no competing interests.

