## [Peer Review File · Nature Communications]

Reviewers' comments:

Reviewer #1 (Remarks to the Author):

The manuscript by Valderrama et al describes what is claimed to be a new method to describe the dynamics of crowd behavior. The method purports to derive critical parameters for how individuals interact based on global distributions. The work, as presented, is difficult to follow, poorly referenced and as such, not yet ready for publication. I list below three main issues with the manuscript.

1. The work is not well placed in the huge literature of spatial point analysis. Much is known in the field and yet reading this manuscript, it seems that all being invented here. Many of the ideas here are formulated in Diggle, "Statistical Analysis of Spatial Point Patterns", Arnold Publishers. Many approaches are described in this book as well as the very many following papers that reference it, which infer individual interactions based on bin-variance. I am having trouble finding what is specifically new here.

2. The use of fruit flies is potentially useful, but much is also known about their spatial distributions. They are much more complicated than as described here. See, for instance "Social structures depend on innate determinants and chemosensory processing in *Drosophila*" PNAS 2012. A very similar study is carried out in this PNAS paper but is overlaid with more intricate network analysis. The high temperature (50degC) and long observation periods (many hours) are also problematic. The flies are either being cooked, starved or dying of thirst. What is being examined is an unhealthy population that is also changing in time. It would be far more preferable to use more physiological conditions, which are well described (see PNAS paper), so that the general behavior is constant over time.

3. This is a very difficult manuscript to follow. The first main paragraph describes an agent based model in significant detail - which has nothing to do with the actual work. The figures and text are hard to follow. The authors should come up with one general parameter at the start and follow it all the way through. Instead they switch from vexation, to frustration to free-energy, to DFFT and onto DFT. If they were to pick one, clearly describe and defend its value at the start, their case would be so much easier to defend.

Overall, this is work that could be great but in its current form is not acceptable.

Reviewer #2 (Remarks to the Author):

In this paper, the authors take a new approach to describing the dynamics of crowds and other aggregations of organisms by adapting ideas from hard condensed matter theory and statistical mechanics, introducing what they term a density functional theory for these systems. Although I can see this idea upsetting some purists, as the connection to traditional density functional theory is certainly only an analogy, in my opinion the introduction of new ideas like this is exactly what is needed in this field rather than a further re-hashing of the same old tired Vicsek-style models. Ultimately, time will tell whether this new density functional theory of crowds is useful; but as an intriguing, novel idea, I am very supportive of publication, and a high-profile venue like Nature Communications will spread the ideas widely.

I have only a few comments that the authors may want to consider in revisions.

First, unless I missed it, I believe that the acronym "DFFT" is never actually defined (it appears first on p. 5). The title of the paper makes it clear what this acronym stands for, but it would be nice to include an explicit definition in the text.

Second, in their introduction, the authors make an appeal to thermodynamics and statistical mechanics to argue that emergent features of an aggregation of animals or other biological agents may be relatively insensitive to the details of the underlying agents and interactions. I completely agree with this notion, but the authors are certainly not the first to argue for this point of view. They cite only two theory papers, neglecting recent experimental advances along these lines. I might suggest including more references here, such as work from Cavagna and Giardina in Rome (e.g., *Nature Physics* 13, 914, 2017), Hu and Fernandez-Nieves at Georgia Tech (e.g., *Nature Materials* 15, 54, 2016), or Ouellette at Stanford (e.g., *Phys Rev Lett* 119, 178003, 2017). In addition, some discussion of theoretical work arguing that things are different for active systems (e.g., Solon et al., *Nature Physics* 11, 673, 2015) may also be warranted.

Finally, somewhat relatedly to my last point, I was surprised to see the authors define a free energy on p. 5, given that their system is out of equilibrium. Can the authors include some commentary on the validity of this assumption, and the interpretation of what this free energy would mean?

Reviewer #3 (Remarks to the Author):

In the present paper the authors introduce a novel approach to comprehend, describe and predict the intra-group movements of agents subject to two kinds of influences: (i) environmental effects, and (ii) the influence of their mates. In order to do so, the authors first construct a plausible agent-based model for a situation in which various locations are characterized by values describing their desirability (representing the environmental effect), furthermore, the agents are effected by the density of their mates as well, captured by a "frustration function" being correlated to the mood of the group. The exemplar situation is a crowd at a political rally, where positions close to the stage are more preferred while overcrowded places are avoided. Then, based on this agent-based model in which the emergent collective behaviour is inferred from the local (inter-agent) rules, the authors derive a corresponding "top-down" model in which functions deduced from the local interactions are able to describe the dynamics of the population density. The big advantage of this derived model, according to the authors, is that it is appropriate "to infer the rules for mass behaviours directly from observations of local crowd density and to quantitatively predict mass behaviour under new circumstances" (see middle of the Abstract). Furthermore, this new "formulation dramatically reduces the complexity of the system description". (top of page 4) Finally, the theoretical results are compared to experimental results. As a whole, I find the results convincing and novel, and I also agree with the claims that it might open the way for developing new methods predicting crowd distributions under various conditions and situations.

My biggest concern regarding the manuscript is that the way in which the presented study fits into the field of collective motion is blurred. According to the Abstract (2nd sentence), "Current agent-based models of collective motion can reproduce many behaviours, ranging from random milling to flocking and schooling, but often must postulate difficult to validate rules for agent interactions with each other and their environment". Based on such an opening, one would expect a model for collective motion (that is, in which the centre of mass of the group moves), without the postulation of "difficult to validate rules for agent interactions with each other and their environment". In contrast, the groups under study are steady in the sense that their centre of mass moves only slightly, that is, within the borders of the group, and also, difficult to validate rules are utilised, for example the one defining the probability with which an agent moves from location x to x' ($1/(e^{\Delta H} + 1)$, page 2).

Accordingly, I suggest to include a few sentences clarifying that the presented model gives account for the intra-group rearrangement of the agents within a group, constituting a special sub-field of collective motion.

I would also welcome a paragraph about the authors' views regarding the possible extensions/modifications of the suggested approach for describing general collective motion scenarios – that is, when the group as a whole moves. I guess it is attainable, since those movements are also resultant of environmental influences (location of food, predators, etc.), and interactions among agents (alignment, collision-avoidance, etc.). (In order to achieve such a generalisation, maybe a new aspect would have to be introduced, giving account for the personal motivations, such as hunger or fear?)

However, as mentioned above, I find the research novel and convincing with important possible applications, primarily regarding human gatherings such as attending political, sport or religious events. Accordingly, I support the publication of the manuscript with the above mentioned modifications.

Anna Zafeiris

We wish to thank the reviewers for their time and careful reading of our original manuscript. We are particularly grateful for their valuable insights, which have resulted in a significantly improved manuscript that we hope is now suitable for publication in *Nature Communications*.

Specifically, we were very pleased to see that all three reviewers find great promise in our work. Reviewer 2 finds our work “an intriguing, novel idea” and is “very supportive of publication”, and Reviewer 3 states that our results are “convincing and novel” and “might open the way for developing new methods predicting crowd distributions under various conditions and situations”. Finally, Reviewer 1 states “this is work that could be great” but then adds “but in its current form is not acceptable” and then enumerates three main areas of concern.

Below, we give a detailed point-by-point response to all issues raised by the reviewers. Here, we briefly summarize our responses to the deeper concerns of Reviewer 1, who enumerates three major areas for improvement: (1) that the reviewer is “having trouble finding what is specifically new here”, (2) that we oversimplify fly behavior and that the conditions of our experiments are not ‘physiological’, and (3) that the presentation is confusing. In regard to area (1), below we detail the novelty of our work as compared to the prior art in point-pattern analysis and illustrate how density-functional theory (DFT) enables us to overcome past shortcomings and actually, for the first time, predict behavior in entirely new environments. In regard to (2), we explain how our goal is to establish that, indeed, despite the complexity of the behavior, we can make accurate predictions of overall population behavior without fully understanding that behavior. Also, we explain how our use of perturbed conditions is intentional - we want to elicit changes in behavior - and follows standard approaches from the literature for this purpose. Finally, to address (3), we have made significant changes to the manuscript to clarify the above issues, as well as our overall line of reasoning.

Below, we repeat all of the comments from each reviewer, interspersing both our responses to the concerns raised and a detailed description of the resulting changes to the manuscript. We look forward to hearing again from the editors and referees.

Reviewer #1

The manuscript by Valderrama et al describes what is claimed to be a new method to describe the dynamics of crowd behavior. The method purports to derive critical parameters for how individuals interact based on global distributions. The work, as presented, is difficult to follow, poorly referenced and as such, not yet ready for publication. I list below three main issues with the manuscript.

The reviewer proceeds to break down these points below. After each of his more detailed comments, we give our detailed responses. Overall, our response has been to improve the discussion in the manuscript by better explicating the connections of our DFT approach to the prior art in spatial point-pattern analysis through the inclusion of additional references and

changes to the text as necessary. Because the other two referees did follow the manuscript quite well, we believe that the overall line of reasoning was sound, and we are hopeful that by better contextualizing our work for those familiar with spatial point-pattern analysis, the presentation will now be clear to all readers.

1. The work is not well placed in the huge literature of spatial point analysis. Much is known in the field and yet reading this manuscript, it seems that all being invented here. Many of the ideas here are formulated in Diggle, "Statistical Analysis of Spatial Point Patterns", Arnold Publishers. Many approaches are described in this book as well as the very many following papers that reference it, which infer individual interactions based on bin-variance. I am having trouble finding what is specifically new here.

Reviewing our manuscript in light of Referee 1's response, we agree in retrospect that we should have better contrasted our work from the field of work presented in *Diggle*, as there are quite clear, significant, and important differences. There are a number of ways to see that this is the case.

First, our work is the first to bring the powerful techniques of density-functional theory from the field of statistical physics to bear on the behavior of mobile crowds. Many of the concepts and analyses described in *Diggle* have direct analogues in statistical physics, including study of the various spatial correlation functions and expansion of the overall joint probability distribution in terms of a Poisson process, pair correlations, triplets, etc. Density-functional theory, which garnered Walter Kohn his share of the 1998 Nobel prize in Chemistry, takes a different, and extremely powerful, tack. Rather than focussing on N-body correlation functions with their attendant computational complexity, density-functional theory allows one to reach exact conclusions on the behavior of a statistical mechanics system (whether classical or quantum), regardless of the complexity of the underlying interactions, by focussing *solely* on the local density $n(r)$ as the central descriptor, with a single *universal* (i.e., applicable to all possible external environments) functional $F[n(r)]$ of the density and a single function of space $V(x)$ determining the behavior of the system. The power of DFT is the recognition that the universality of the functional $F[n(x)]$ allows for making *predictions in new environments*, a step not taken in the field of spatial point-pattern analysis.

Diggle's work itself strongly supports the view that we have done something novel in the field of point distributions, particularly in that he reports that the statistic that we employ here had been abandoned as a fruitful line of inquiry. As Reviewer 1 points out, it is true that there are significant connections between our work and spatial point-pattern analysis. For example, our "bin count distribution" (one of our central quantities describing the statistics of the local density) corresponds to the "quadrat count data" described in *Diggle*, and that our "neutral interaction" (no interaction/zero frustration) case corresponds to the "single parameter Poisson distribution". It is also true that workers in spatial point-pattern analysis had noticed that quadrat distributions can vary from Poisson statistics due to interactions, as our DFT analysis also predicts. In the

prior art, the dominant approach was to then attempt to recognize the form of the quadrat distribution and thereby characterize the nature of the statistics of the system under study, and, in some cases, to attempt to infer the nature of the underlying process. On p. 41-42 of the 2013 edition of the book, Diggle himself writes

Early work on the analysis of quadrat count data concentrated on the development of more general families of discrete distributions than the single parameter Poisson, especially with a view to modelling aggregated patterns. See, for example, Evans (1953) or Douglas (1979). The story of the rise and fall of these so-called “contagious distributions” as tools for the analysis of spatial data is of some historical interest because it shows how attempts to model observed data through discrete *distributions ultimately founded* on their failure to respect the underlying setting of spatial point processes. [underlining added, emphasis in the original]

In contrast, our approach is to take the quadrat statistics, as measured for a given population type, *without bias toward any preconceived form of the distribution*, and to use these statistics to predict crowd behavior under entirely new conditions. Indeed, the above quote indicates that, as recently as 2013, pursuing the quadrat statistic was regarded as a dead end. Certainly nothing in *Evans* or *Douglas* comes anywhere close to the capability we demonstrate in predicting, from past experimental data, the results of entirely new experiments on crowds in new environments — one of the primary strengths of our technique.

To Reviewer 1’s point, certainly the original manuscript did not contrast our approach against the prior art with sufficient clarity. We also recognize, in retrospect, that we did not sufficiently bridge the communication gap between the fields of physics and statistics. Therefore, we have modified the manuscript in two substantial ways. First, we now do a better job contrasting our work against statistical point analysis, and second we now call out the terms in our presentation with corresponding analogues from statistical point analysis along with appropriate references to *Diggle’s* review and to original source material where more appropriate.

Concepts from the prior art appearing in our work include the quadrat count (bin count), intensity (vexation= $-\log(\text{intensity})$), complete spatial randomness (neutral interactions), and contagious quadrat distributions. Among the new concepts we introduce are (a) a fully general, flexible, and detailed notion of frustration that can be used predictively in entirely new environments with different population sizes, (b) a notion of vexation (intensity) that is independent of the size of the population over multiple orders of magnitude, and (c) the focus on quantifying behavior directly from experiments in order to predict density-distributions in new environments.

Changes to the manuscript: We have incorporated the following changes to the new manuscript to better place our work in context with prior work as described above:

1. To alert the readership to prior work studying the statistics of local densities, we add in the following sentence with the corresponding term “quadrats” explicitly mentioned and referenced on page 4:

Since we cannot measure the function $n(x)$ directly in experimental crowds, we instead consider discrete counts of individuals within equal area bins (quadrats)²⁴.

2. To make the connection with terminology in the point-pattern analysis literature, we use the phrase “*complete spatial randomness*” instead of “independent” on page 4 when describing a random non-interacting crowd.

In particular, ~~if there are no interactions~~ in the case of neutral interactions ($f_N = 0$), the ~~fluctuations~~ bin counts will be single-parameter Poisson distributed, as expected for an experiment counting so-called ~~independent~~ completely spatially random events²⁴.

3. To acknowledge a correspondence between our concept of vexation and the statistical parameter “intensity”, which both relate to the Poisson parameter, we add in the following clause and reference on page 4:

From the mean of these distributions one can extract an effective v_b (Fig. 1b,c) or logarithm of the so-called intensity²⁴ that can arise either from actual preferences for particular locations or from other kinetic interactions with the environment, such as slowing down near barriers.

4. To acknowledge other statistics-based approaches that also derive a modified Poisson form for describing local densities, we add the following referenced statement on page 4:

For example, so-called contagious distributions, which correspond to attractive interactions and show increased variance-to-mean ratios, have been observed^{24,28,29}.

5. To further contrast our work with the point pattern analysis approaches, we add a clause in the following statement underscoring the fact that our measurement of interactions between individuals does not attempt to fit to a preconceived functional form, as is done in the field of point-pattern analysis:

Finally, from distortions off of the Poisson form, we can determine an effective function f_N , without assuming any particular functional form, that describes any local interaction, attractive or repulsive.

6. To acknowledge previous work on the field of spatial-point pattern analysis we included the relevant terminology regarding Poisson-distributed counts in the caption of figure 1:

For neutral interactions, we expect complete spatial randomness leading to Poisson distributed counts within each bin.

7. To acknowledge the fact that modified Poisson distributions emerge from non-interacting quadrat counts when interactions are included, we have added a reference to Diggle 2013 in the caption of figure 1 (d-f):

attractive interactions are thus reflected in the deviation of the probability distribution from the Poisson form²⁴.

2. The use of fruit flies is potentially useful, but much is also known about their spatial distributions. They are much more complicated than as described here.

We are in total agreement with the reviewer that it is common knowledge that many stimuli, including heat, olfactory and light, can shift the spatial distributions of the flies. Our purpose here, however, is not *per se* to discover new facts about how flies distribute, but rather to show that one can predict precisely how they will distribute under new circumstances as well as quantify changes in their behavior. Our purpose is also to show that one can accomplish both goals from simple observations of the overall density, rather than, for example, of the complex underlying social network.

Changes to the manuscript:

- 1) To clarify our intent, we now include citations to the fly spatial behavior literature in the revised manuscript, along with the explicit statement that our use of these techniques is to test the utility of the concepts of vexation and frustration (which we extract from observations of the density):

It is well known that flies exhibit complex spatial preferences^{30,31} and social behaviors^{32,33}. Here we seek to determine whether a large crowd of individuals with such complex behaviors indeed can be described within our vexation and frustration framework.

See, for instance “Social structures depend on innate determinants and chemosensory processing in *Drosophila*” PNAS 2012. A very similar study is carried out in this PNAS paper but is overlaid with more intricate network analysis...

Certainly, a great deal of work has been applied to understanding the social behavior of *Drosophila* including the article the reviewer mentions. We were indeed previously aware of this work and its nature — it was cited in the review article we reference in our original manuscript (“One, Two, and Many,” J. Schneider, 2012, cited in the updated manuscript as reference 32). Although both the PNAS paper and our experiment use data from groups of *Drosophila*, our method of analysis and the primary focus of our work are drastically different. The network analysis mentioned by the reviewer defines contacts between two flies in a group based on the relative positions and postures between those flies in order to develop a social network model. This method for observing interactions does not apply directly to a variety of systems and is limited since it only includes pair interactions when flies are face-to-face.

We, however, seek a general approach that is applicable to a wide variety of crowd-like systems, rather than a detailed understanding of *Drosophila* behavior. Our first aim is to make *predictions* of overall population distributions *directly* from observations of the overall population. The power of DFFT is that we do not need to know all of the underlying social network complexities, and yet we are still able to make highly accurate predictions in entirely new environments. Our second aim is to test the power of our frustration functional to *measure* the changing behavior of the flies and thus to demonstrate that our density dependent metric, which in principle is easily extensible to other systems, can indeed quantify distinct behavioral modes.

Changes to the manuscript:

- 1) We add on p. 5 a sentence contextualizing our work with other studies and clarifying that we use *Drosophila* as a model system to test our technique, and that our technique should apply more broadly to crowds in general. We have also added an explicit reference to the paper mentioned by the referee in this sentence of the manuscript:

It is well known that flies exhibit complex spatial preferences^{30,31} and social behaviors^{32,33}. Here we seek to determine whether a large crowd of individuals with such complex behaviors indeed can be described within our vexation and frustration framework.

The high temperature (50degC) and long observation periods (many hours) are also problematic. The flies are either being cooked, starved or dying of thirst.

Our use of temperature and starvation/dehydration are *intentional* perturbations to the experiment, but they are not unusual and align with general protocols found in the walking fly literature. Zars et. al. (*Journal of Comparative Physiology A*, 2006), use hot and cold temperature preferences to assess spatial learning. Starvation is commonly used to probe foraging behaviors: Bell et. al. (*Animal Behavior*, 1985) starve flies for up to 48 hours, many times longer than we do. Lastly, Ji. et. al. (*PLoS one*, 2015) use dehydrated flies to elicit hygrotactic behaviors. We clarify these points for the reader by including references to the pertinent literature in the revised manuscript.

Changes to the manuscript:

- 1) To make more explicit the connections with the temperatures and deprivation times found in previous literature, we now include the references mentioned above to our main text on page 5 and our Methods section on page 15.

To explore a variety of behaviors, we use arenas of different shapes³⁰ and apply heat gradients³⁴ across the arenas to generate different spatial preferences. (page 5)

...

To probe the changing fly behaviors shown in figure 4, we track the flies for up to 9 hours before flies begin to die from ~~consumption~~ deprivation^{45,46}. (page 15)

...

*To elicit different behaviors and location preferences with the same population of flies, we apply a heat gradient **to generate an avoidance behavior³⁴** starting at 20 minutes after being introduced to the chamber. (page 15)*

- 2) To more explicitly describe the experimental conditions for the heat-gradient experiment, we add the following statement to the Methods section on page 14.

*Chamber temperature is measured for two locations using a contact thermometer to ensure **no more than 2 degrees Celsius drift and consistent temperature gradients** consistency of heating **between trials**. ~~For each chamber, we choose the temperature gradient by turning up the heat to the point where flies do not~~*

~~die, typically from 30-50 degrees Celsius.~~ *We heat one side of the chamber to temperatures between 40-50 degrees Celsius³⁴. The opposing side of the chamber is connected to a heat sink and kept at temperatures between 25-35 degrees Celsius. We find that the resulting temperature gradient drives a strong avoidance behavior for the hotter wall while avoiding fly death as the flies avoid the high-temperature region.*

What is being examined is an unhealthy population that is also changing in time. It would be far more preferable to use more physiological conditions, which are well described (see PNAS paper), so that the general behavior is constant over time.

We note that our aim in the use of the heat and deprivation perturbations to the flies is to demonstrate the power of our DFFT technique as a quantitative metric for *behavior* in general crowd systems, rather than to characterize behaviors of flies under a particular ideal laboratory condition. Instead, we seek to quantify the differences between measurements of the population behavior under different conditions through observations of the overall local density *alone*. We thus, for this first application of DFFT to crowds, intentionally use significantly modified conditions to elicit different behaviors for our measurements.

In regard to the evolving behavior in time, a common approach in walking fly experiments is to sample behaviors for a certain time window and then verify that the sought after behaviors are staying constant. For example, Berman et al. in oral presentations frequently confirm the stability of their behavioral ethograms (*Mapping the stereotyped behaviour of freely moving fruit flies*, Berman et. al. 2014) over their observation time windows. To test whether the behaviors we observe are changing over our 10 minute time windows, we accordingly compare the probability distributions from the first half of the window with the second half.

Changes to the manuscript:

- 1) Again to ensure that our aims are clear to the readers, we have included the statement mentioned several times above, namely:

It is well known that flies exhibit complex spatial preferences^{30,31} and social behaviors^{32,33}. Here we seek to determine whether a large crowd of individuals with such complex behaviors indeed can be described within our vexation and frustration framework.

- 2) In addition, we now add the following statement to our Methods section on page 15 that describes how we ensure that the behavior we observe is not changing during our observation windows:

To test whether fly behavior is changing over our standard 10 minute time windows, we compare the probabilities, $P_b(N)$, from the first 5 minutes of the window with the last 5 minutes and find that they are consistent. The only exception to this is during the very first 5 minutes after the flies are introduced

into the chamber as they become oriented to their new environment that we do not include in our analysis.

3. This is a very difficult manuscript to follow.

We recognize that it is our responsibility to prepare a manuscript that is clear and easy to follow for researchers from a wide variety of fields. Based on Reviewer #1's comments, we recognize that the original manuscript did not meet this criterion, and have made a number of changes accordingly. Before discussing those changes, we would like to note that neither Reviewer 2 nor 3 expressed any concerns regarding the clarity of the line of argument in the manuscript. Further, they both raised a number of questions which clearly indicate a good understanding of the work. On these lines, Review #3 included a near-perfect summary of our line of argument in its first and second paragraphs, addressing, for example, one of the main points of confusion raised by Reviewer 1 (the role played by the agent-based model as discussed below). Nonetheless, it is clear from Reviewer 1's comments that the original manuscript would be confusing for some researchers. We are very grateful to Reviewer 1 for the careful reading of the original manuscript and informing us of this issue.

In particular, we recognize that the manuscript did not well contextualize our work against spatial-point pattern analysis. We hope, that with the changes we have made to the manuscript to better explain our work for those familiar with the terminology from spatial-point analysis, that Reviewer 1 will now find the presentation of the line of reasoning in the revised manuscript to be acceptable. We also have made significant additional changes in response to the reviewer's further comments on these lines, as we detail below.

The first main paragraph describes an agent based model in significant detail - which has nothing to do with the actual work.

It is in fact not true that the paper makes no use of the model described in the first main paragraph. Rather, as Reviewer #3 points out, our approach is to

"first construct a plausible agent-based model for a situation in which various locations are characterized by values describing their desirability (representing the environmental effect), furthermore, the agents are effected [sic.] by the density of their mates as well, captured by a "frustration function" being correlated to the mood of the group."

so that

"Then, based on this agent-based model in which the emergent collective behaviour is inferred from the local (inter-agent) rules, the authors derive a corresponding "top-down"

model in which functions deduced from the local interactions are able to describe the dynamics of the population density.”

In summary, the model serves as a probe of the wide range of behaviors that lead to the family of probability distributions for the local density described in the manuscript. Since the other reviewers did not have concerns regarding how the model connects to the rest of the work, we decided not to make significant modifications to this section of the paper. Nonetheless, Reviewer #1 is right in that this connection can be stated with more clarity. As such, we changed the way in which the agent-based model is introduced to stress its connection to the body of work developed in the later sections of the manuscript.

Changes to the manuscript:

We have made the following modifications in relation to the above mentioned points.

- 1) To better prepare the reader for the purpose of defining our agent based model, we add in the following sentences on page 2 of our manuscript.

Accordingly, our strategy is to begin with a family of models that roughly capture the "microscopic" behaviors of individuals in a crowd. We do this, not because we are interested directly in individual behaviors, but rather because we are interested in the generic "macroscopic" behaviors that emerge in crowds en masse.

- 2) To emphasize the purpose of introducing the agent based model (to probe a wide class of behaviors that roughly follow similar sets of rules) we introduced the following sentence in page 3 of the manuscript.

Again, our purpose here is not to develop such a model in detail, but rather to explore the top-level, global behaviors that emerge from this class of models, which we conjecture should apply to crowds more generally.

The figures and text are hard to follow.

As we have detailed above and below, we have made significant changes to the text, which we believe make our line of argument much easier to follow. Reviewer 1 does not specify what precisely about the figures may be hard to follow, and we note that no other Reviewer made mention of the figures. Nonetheless, we have taken the opportunity to make some small improvements to the figures and captions.

Changes to the manuscript:

We made modifications to figure 1 and 3 to make them more clear.

- 1) The top right plot for the vexation in figure 1 was modified to mirror more closely the image for the simulated crowd in the top left.
- 2) We correct a typo in figure 3b by changing the " v_i " to " v_b " for the vexation label.

The authors should come up with one general parameter at the start and follow it all the way through. Instead they switch from vexation, to frustration to free-energy, to DFFT and onto DFT. If they were to pick one, clearly describe and defend its value at the start, their case would be so much easier to defend.

It would not be possible to focus our work on any one of these items. The first two items (vexation and frustration) are essential parameters describing the overall behavior of the crowd. We must present a joint analysis for vexation and frustration to capture all (environmental and inter-agent) interactions relevant to the crowd behaviors we describe. As such, we cannot drop either concept from our analysis.

The next item listed by the Reviewer, free energy, does not play a key role in our work. We introduced “effective free energy” (now “pseudo free energy”), not as a new concept, but as a mnemonic for a formula arising from our analysis ($-\ln N! P(N)$). To help clarify this, the manuscript now explicitly refers to its use as a mnemonic.

Finally, the last two items listed by the Reviewer, DFT and DFFT, are not parameters but refer to the type of analysis we carry out. We agree with Reviewer #1 in that switching between density-functional theory (DFT) and density-functional fluctuation theory (DFFT) may lead to confusion, especially if these acronyms and concepts are not properly identified when they first occur in the manuscript. However, we must include mention of DFT to put the work in its proper context because DFT clearly is the inspiration for our work. We must also mention DFFT because what we actually do is somewhat different from traditional DFT analyses. Accordingly, we have taken steps in the new manuscript to ensure that the general reader understands our use of these terminologies.

Changes to the manuscript:

To address the lack of introduction of the relevant terminology in our paper (DFT and DFFT), we made the following modifications to the manuscript:

- 1) “Free energy” mnemonic:

To validate our description and quantify the vexations and frustrations, we plot what we call as a mnemonic the “effective pseudo-free energy” $-\ln(N! P_b(N)) = (v_b N + \ln(z_b)) + f_N$ versus N in figure 2f.

- 2) Now, the acronym DFT is defined in the abstract

Underlying this approach is a statistical framework employing concepts from classical density-functional theory (DFT).

- 3) A sentence that introduces the acronym DFFT has been added to page 4.

Because this reduction in the number of variables is the result of transitioning to a local-density description as in classical density-functional theory, but now with the modification that interactions are inferred from density fluctuations, we call our approach density-functional fluctuation theory (DFFT).

Overall, this is work that could be great but in its current form is not acceptable.

With the extensive changes detailed above, we are optimistic that the revised manuscript is now acceptable.

Reviewer #2

In this paper, the authors take a new approach to describing the dynamics of crowds and other aggregations of organisms by adapting ideas from hard condensed matter theory and statistical mechanics, introducing what they term a density functional theory for these systems. Although I can see this idea upsetting some purists, as the connection to traditional density functional theory is certainly only an analogy, in my opinion the introduction of new ideas like this is exactly what is needed in this field rather than a further re-hashing of the same old tired Vicsek-style models. Ultimately, time will tell whether this new density functional theory of crowds is useful; but as an intriguing, novel idea, I am very supportive of publication, and a high-profile venue like Nature Communications will spread the ideas widely.

I have only a few comments that the authors may want to consider in revisions.

First, unless I missed it, I believe that the acronym "DFFT" is never actually defined (it appears first on p. 5). The title of the paper makes it clear what this acronym stands for, but it would be nice to include an explicit definition in the text.

Changes to the manuscript:

- 1) We now explicitly define our DFFT acronym by adding the following statement on page 4 of our manuscript.

Because this reduction in the number of variables is the result of transitioning to a local-density description as in classical density-functional theory, but now with the modification that interactions are inferred from density fluctuations, we call our approach density-functional fluctuation theory (DFFT).

Second, in their introduction, the authors make an appeal to thermodynamics and statistical mechanics to argue that emergent features of an aggregation of animals or other biological agents may be relatively insensitive to the details of the underlying agents and interactions. I completely agree with this notion, but the authors are certainly not the first to argue for this point of view. They cite only two theory papers, neglecting recent experimental advances along these lines. I might suggest including more references here, such as work from Cavagna and Giardina in Rome (e.g., Nature Physics 13, 914, 2017), Hu and Fernandez-Nieves at Georgia Tech (e.g., Nature Materials 15, 54, 2016), or Ouellette at Stanford (e.g., Phys Rev Lett 119, 178003, 2017). In addition, some discussion of theoretical

work arguing that things are different for active systems (e.g., Solon et al., Nature Physics 11, 673, 2015) may also be warranted.

Changes to the manuscript:

3) We have added the suggested references in the introduction, as suggested (page 2):
Remarkably, these emergent behaviors are often insensitive to the detailed nature of the underlying interactions. Here, we pursue the hypothesis that a similar scenario emerges in the study of large ~~populations~~ crowds¹⁸⁻²² so that behaviors arising from generic agent-based models can be predicted using a top-down approach.

4) In accordance with the reviewer's recommendation, we add an additional clause (page 2) acknowledging the differences inherent in applying statistical physics and thermodynamic approaches to active systems alongside the recommended reference:

This tack is not a priori obvious since active systems do not possess a fixed energy, their temperature is ill-defined, and there are no obvious equilibrium states²³. Nonetheless, we shall see that mathematical equivalents of free energy, the Hamiltonian, and equilibrium states arise naturally from plausible models of crowd behavior.

Finally, somewhat relatedly to my last point, I was surprised to see the authors define a free energy on p. 5, given that their system is out of equilibrium. Can the authors include some commentary on the validity of this assumption, and the interpretation of what this free energy would mean?

We are not claiming that the quantity that is defined in the main text has the properties of a free energy in the strict, physics sense. Instead, the terminology is introduced to elicit the similarity between the quantity that arises in our analysis, the logarithm of a probability distribution, and the usual formula for calculating free energies starting from the canonical ensemble.

Changes to the manuscript:

1) To clarify our use of the notion of free energy, we switch the qualification of free energy from an "effective" free energy to "pseudo" free energy wherever it is mentioned (page 5 and the figure and text for figure 2):

-f, The ~~effective~~ pseudo-free energy, $-\ln(N!P_b(N))$, for eight representative bins.

-Frustration functional, f_N , obtained from collapse of the ~~effective~~ pseudo-free energies for all 48 bins upon removal of the Poisson contributions.

-h, Vexation for each bin as measured from the Poisson contributions to the pseudo-free energies.

-To validate our description and quantify the vexations and frustrations, we plot the ~~effective~~ pseudo-free energy $N!P_b(N) = v_b N + \ln(z_b) + f_N$ versus N in figure 2f.

-Methods: To recover the frustration-vexation probability form analyzed throughout the text, we now follow the standard Statistical Mechanics approach

of defining an effective pseudo-free-energy functional by integrating out internal degrees of freedom.

Reviewer #3

In the present paper the authors introduce a novel approach to comprehend, describe and predict the intra-group movements of agents subject to two kinds of influences: (i) environmental effects, and (ii) the influence of their mates. In order to do so, the authors first construct a plausible agent-based model for a situation in which various locations are characterized by values describing their desirability (representing the environmental effect), furthermore, the agents are effected by the density of their mates as well, captured by a “frustration function” being correlated to the mood of the group. The exemplar situation is a crowd at a political rally, where positions close to the stage are more preferred while overcrowded places are avoided.

Then, based on this agent-based model in which the emergent collective behaviour is inferred from the local (inter-agent) rules, the authors derive a corresponding “top-down” model in which functions deduced from the local interactions are able to describe the dynamics of the population density. The big advantage of this derived model, according to the authors, is that it is appropriate “to infer the rules for mass behaviours directly from observations of local crowd density and to quantitatively predict mass behaviour under new circumstances” (see middle of the Abstract). Furthermore, this new “formulation dramatically reduces the complexity of the system description”. (top of page 4) Finally, the theoretical results are compared to experimental results.

My biggest concern regarding the manuscript is that the way in which the presented study fits into the field of collective motion is blurred.

According to the Abstract (2nd sentence), “Current agent-based models of collective motion can reproduce many behaviours, ranging from random milling to flocking and schooling, but often must postulate difficult to validate rules for agent interactions with each other and their environment”. Based on such an opening, one would expect a model for collective motion (that is, in which the centre of mass of the group moves), without the postulation of “difficult to validate rules for agent interactions with each other and their environment”. In contrast, the groups under study are steady in the sense that their centre of mass moves only slightly, that is, within the borders of the group and also, difficult to validate rules are utilised, for example the one defining the probability with which an agent moves from location x to x' ($1/(e^{\Delta H} + 1)$, page 2). Accordingly, I suggest to include a few sentences clarifying that the presented model gives account for the intra-group rearrangement of the agents within a group, constituting a special sub-field of collective motion.

Here, the Reviewer raises a concern regarding the presentation of the scope of application of our framework and recommends a remedy. We agree with the Reviewer that the opening paragraph raises the expectation that our method could be applied directly to predict collective systems in which the center-of-mass of the crowd is moving significantly, which is indeed not quite true. Accordingly, we have implemented the reviewer's recommended remedy. Specifically, we have made significant changes to a key sentence in the introductory paragraph to focus more directly on the type of systems we actually study, and defer discussion of center of mass motion and flocking to a paragraph in the conclusion on possible future directions. We also add a second sentence to the first paragraph of the main discussion stating explicitly that we are interested in agents as they "rearrange within a crowd."

Changes to the manuscript:

- 1) To better prepare the reader for the systems that we address and the approach we propose, we modify the introductory sentences on page 1, which now reads...
A primary goal of collective population behavior studies is to determine the rules governing the distribution of a crowd, and use these rules to make predictions about future behaviors in new environments. Current agent-based models of ~~collective motion~~ crowds can reproduce many emergent behaviors, ranging from random milling to ~~flocking and schooling~~ swarming, but often must postulate ~~preconceived~~ difficult-to-validate rules for agent interactions with each other and their environment.
- 2) To address applications to flocking systems, we add a note regarding how work would have to be extended to time-dependent density-functional theory to address such systems:
Moreover, by including the local current density ("flow") in the functional, such approaches may even be able to describe crowds where correlated subgroups move with different local velocities, such as in flocks of birds.
- 3) To put proper emphasis on rearrangements within a group/crowd, we have added the sentence below to p.2
Accordingly, our strategy is to begin with a family of models that roughly capture the "microscopic" behaviors of individuals as they rearrange within a crowd. We do this, not because we are interested directly in individual behaviors, but rather because we are interested in the generic "macroscopic" behaviors that emerge in crowds en masse.

I would also welcome a paragraph about the authors' views regarding the possible extensions/modifications of the suggested approach for describing general collective motion scenarios – that is, when the group as a whole moves. I guess it is attainable, since those movements are also resultant of environmental influences (location of food, predators, etc.), and interactions among agents (alignment, collision-avoidance, etc.). (In order to achieve

such a generalisation, maybe a new aspect would have to be introduced, giving account for the personal motivations, such as hunger or fear?)

We are indeed very interested in these general scenarios of collective motion. However, the extension to mobile crowds entails several new aspects (as anticipated by the Reviewer) which require significant modifications to our current formalism. These are items which are beyond the scope of the present study. Consequently, we hesitate to provide detailed conjectures in the present manuscript beyond a rough sketch of possible future avenues of research. In response to the Reviewer's suggestion, we now separate out such possible extensions into a new paragraph and sketch how future researchers could attack such questions.

To answer the Reviewer's questions directly, we provide here some additional details which fall outside the scope of the present work. There are two types of center of mass motion at issue. The first behavior is one in which the center of mass of the population moves, as in herding and swarming, but there is not *per se* a velocity correlation built directly into the dynamics. For these situations, direct extension to time-dependent density-functional (TDDFT) within the *adiabatic* approximation should suffice. In the phenomenologically richer case where there is velocity correlation, such as Vicsek-style flocking, more in-depth extensions of TDDFT, such as inclusion of the local current density \mathbf{J} ("flow") into the time-dependent functional should, we feel, suffice to give a good description of such systems. Finally, shifting motivations such as hunger or fear would be accounted by modifications to both the vexation and frustration functionals, as we already exemplify by the changing frustrations for our flies as they become hungry/thirsty toward the end of the experiment. Vexation too would be modified due to hunger, where locations with food would develop far lower vexation values as a result. Proper description of the resulting motion of the crowd overall would then require the technical extensions to our approach as described above.

Changes to the manuscript:

- 1) To better describe the work's potential application to other systems of interest in collective behavior, we now separate out our discussion of possible theoretical extensions into a separate paragraph and add to that paragraph a discussion of the possibility of extensions along the lines suggested by the referee (p. 10):

There are a number of directions in which the formal framework suggested here can be extended, paralleling developments from the traditional density-functional theory literature. Extensions to time-dependent DFT methods (TDDFT)^{36,37} would enable the prediction of situations in which crowds gather and disperse in response to changes in the environment. This approach would also apply to situations in which the center of mass of the entire group is moving as whole, such as in herd migration and bacterial and insect swarming. Moreover, by including the local current density ("flow") in the functional, such approaches may even be able to describe crowds where correlated subgroups move with different local velocities, such as in flocks of birds. Likewise, extensions to

multicomponent DFT³⁸ would enable corresponding predictions and observations in crowds composed of distinct groups exhibiting interactions such as inter-group conflict, predator-prey relations, or mating behavior.

REVIEWERS' COMMENTS:

Reviewer #1 (Remarks to the Author):

I think that the authors have made a heroic attempt to address my concerns. I still have some reservations about the application to flies, but also concede to their answers to my concerns. This is a novel method and could have very broad applications. I suggest two small changes: please identify the wild type strain of flies used (CantonS?) and change the 'drosophila' in all of the references to 'Drosophila'.

Reviewer #2 (Remarks to the Author):

I am satisfied with the revisions the authors have made, and recommend publication.

Reviewer #3 (Remarks to the Author):

The authors have answered all my comments and question in a satisfactory way and have modified the paper accordingly. I consider the new manuscript as being appropriate for publication.

We wish to thank the reviewers for their time and reading of our revised manuscript. We are very pleased that, pending “two small changes” suggested by Reviewer #1, all three reviewers now support publication of our manuscript in *Nature Communications*. We are grateful for the final two points of clarification raised by Reviewer #1, to identify the strain of flies used and to correct the capitalization of *Drosophila* in the references, both of which we have implemented in the newly revised manuscript.

Below, we give a detailed point-by-point response to all issues raised by the reviewers in the second round of review.

Reviewer #1

I think that the authors have made a heroic attempt to address my concerns. I still have some reservations about the application to flies, but also concede to their answers to my concerns. This is a novel method and could have very broad applications. I suggest two small changes: please identify the wild type strain of flies used (CantonS?) and change the 'drosophila' in all of the references to 'Drosophila'.

We appreciate and adopt both suggested small changes. For our experiments we use an out-bred laboratory stock for which we cannot be sure of a clean genetic background. We now clarify this in our manuscript both in the main text (page 5)

*To test whether this approach applies to actual populations, we consider a model crowd consisting of wild-type male *Drosophila melanogaster* from an out-bred laboratory stock.*

and the Methods section (page 15):

*All experiments were performed 3-15 days post-eclosion using common fruit flies (*D. melanogaster*) from an out-bred laboratory stock reared ~~bred~~ at room temperature on a 12h/12h day-night cycle.*

Finally, we capitalized *Drosophila* in references 31-35 and 45-46.

Reviewer #2

I am satisfied with the revisions the authors have made, and recommend publication.

Thank you.

Reviewer #3

The authors have answered all my comments and question in a satisfactory way and have modified the paper accordingly. I consider the new manuscript as being appropriate for publication.

Thank you.